

# State diagrams to determine tree tensor network operators

Richard M. Milbradt[1*], Qunsheng Huang[1] and Christian B. Mendl[1,2]

**1** Technical University of Munich, School of Computation, Information and Technology,
Department of Informatics, Garching, Germany
**2** Technical University of Munich, Institute for Advanced Study,
Lichtenbergstraße 2a, 85748 Garching, Germany

★ r.milbradt@tum.de

## Abstract

This work is concerned with tree tensor network operators (TTNOs) for representing quantum Hamiltonians. We first establish a mathematical framework connecting tree topologies with state diagrams. Based on these, we devise an algorithm for constructing a TTNO given a Hamiltonian. The algorithm exploits the tensor product structure of the Hamiltonian to add paths to a state diagram, while combining local operators if possible. We test the capabilities of our algorithm on random Hamiltonians for a given tree structure. Additionally, we construct explicit TTNOs for nearest neighbour interactions on a tree topology. Furthermore, we derive a bound on the bond dimension of tensor operators representing arbitrary interactions on trees. Finally, we consider an open quantum system in the form of a Heisenberg spin chain coupled to bosonic bath sites as a concrete example. We find that tree structures allow for lower bond dimensions of the Hamiltonian tensor network representation compared to a matrix product operator structure. This reduction is large enough to reduce the number of total tensor elements required as soon as the number of baths per spin reaches 3.

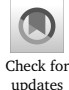

# 1   Introduction

Tensor networks are a description of high-dimensional data with a convenient graphical representation. Their wide applicability leads to the use of tensor networks in a wide range of fields. The specific subclass of tree tensor networks (TTN), introduced in more detail in Section 2, was also successfully applied in many such fields, such as condensed matter physics [1–3] and quantum chemistry [4–7] among others [8–10]. The well-established matrix product structure, also known as the tensor train structure, is a special case of TTN. It occurs if the tree structure of the TTN is simply a chain. Methods for constructing an operator in the matrix product representation, known as the matrix product operator (MPO), have been explored extensively [11–15]. These methods are based on the microscopic Hamiltonian of a quantum system. Notable examples are algorithms using finite state automata that can be converted into MPO tensors [12, 16, 17]. However, analogous approaches using TTNs have not been established so far, i.e., finding a tree tensor network operator (TTNO) given the Hamiltonian of a quantum system. We will base our construction scheme on the data structure called state diagrams [17], which are introduced in section 3. We provide an algorithm that constructs a state diagram equivalent to a Hamiltonian in section 4. A TTNO is then easily obtained from the state diagram. In one dimension, MPOs are the backbone of many algorithms. This includes ground state search [18], time evolution [19, 20] and the simulation of open quantum systems [21–23]. Our results are intended as a step to increase the usability of TTN for the simulation of quantum systems. Some of the methods used in one-dimensional systems, such as the density matrix renormalization group (DMRG) [4–6] and time-evolving variational principle (TDVP) [24, 25], have already been extended to utilize TTN and require the use of Hamiltonians in TTNO form.

# 2   Tree tensor networks

Tensors are a generalization of vectors and matrices in finite-dimensional Hilbert space and can have an arbitrary, finite number of indices. Usually, tensors are considered in the form $M \in \mathbb{C}^{d_1 \times \cdots \times d_k}$, with elements denoted as $M_{i_1,\ldots,i_k} \in \mathbb{C}$ [26]. Notably, extending tensor networks to other sets [27] and to continuous structures is possible [28–30]. However, we restrict ourselves to the definition given above. There exists a canonical graphical notation for tensors in which each index is represented by a line called a leg. Two tensors can be contracted along a leg of equal dimension by summing over the index corresponding to the leg and multiplying the elements with equal index value. The graphical notation reflects this by joining two legs. One can now construct arbitrarily complicated combinations of tensors and contractions. Such combinations are called tensor networks $\mathcal{T} = (\mathcal{M}, C)$, where $\mathcal{M}$ and $C$ are sets of tensors and contractions, respectively. For a thorough introduction to the topic of tensor networks, we refer to various introductory texts available, e.g., [26, 31, 32]. Every tensor network can be

mapped to a graph $\mathcal{G}_{\mathcal{T}} = (V_{\mathcal{T}}, E_{\mathcal{T}})$, where the tensors $\mathcal{M}$ are mapped to the vertices $V_{\mathcal{T}}$ and the contracted legs $C$ are mapped to the edges $E_{\mathcal{T}}$. If the underlying graph of a tensor network is a tree, i.e., without loops, it is called a tree tensor network (TTN) [33]. It is possible for the legs in a TTN to not be contracted. We call these legs physical since they usually correspond to the indices of Hilbert spaces representing actual physical spaces. In this work, we focus on two categories of such TTN. The first is tree tensor network states (TTNS), representing a state in a physical Hilbert space. The other is tree tensor network operators (TTNO), which are operators acting in a physical Hilbert space. Assume we have a TTNS and a TTNO, both with the same tree structure. Then, applying the TTNO to the TTNS, the resulting TTNS will have the same tree structure as our initial state and operator. This effect is desirable as no loops are introduced, which would break the tree structure.

We will now introduce some notation. For a tree $T = (V_T, E_T)$, $V_T$ are the nodes and $E_T$ are the edges. The set of all edges connecting to a node $s \in V_T$ is denoted by $E_T(s)$. Furthermore, the route $\gamma(s, s') = (V_\gamma, E_\gamma)$ between two nodes is the unique smallest collection of nodes and edges that need to be traversed to go from $s$ to $s'$. From this, we can define the distance $d(s, s')$ of the two nodes as $|E_\gamma|$, which defines a metric on the tree. In the case $d(s, s') = 1$, we will also write $s \sim s'$, i.e. $s$ and $s'$ are nearest neighbors. We assume $T$ to be rooted in one of its nodes $r \in V_T$ called root. Consequently, we can define the children of a node $s$ as all $c \in V$, such that $s \sim c$ and $d(c, r) = d(s, r) + 1$. If a node $s$ is not the root, then its parent is the nearest neighbor which is not a child of $s$. Leaves $\ell \in V_T$ are the nodes with no children. The set of all leaves is denoted by $\mathcal{L}(T)$. With respect to the metric, $d$ we define a ball of radius $\chi$ around a midpoint $s \in V_T$ as

$$\mathcal{B}_\chi(s) = \left\{ s' \in V_T \big| d(s, s') \leq \chi \right\}, \tag{1}$$

and its boundary as $\partial \mathcal{B}_\chi(s) = \left\{ s' \in V_T \big| d(s, s') = \chi \right\}$. Finally, for a rooted tree, we define the subtree originating in node $s$ as

$$T_{|s} = \left\{ s' \in V_T \big| s \in \gamma(s', r) \right\}. \tag{2}$$

## 3 State diagrams

In many-body quantum mechanics, the Hilbert space dimension increases exponentially with system size. However, for many models of interest, the Hamiltonian and relevant operators have an exploitable structure. This structure allows the use of finite state automata to store operators, leading to drastically reduced memory requirements [11, 12, 16, 34, 35]. We consider the underlying structure of the automata to be hypergraphs, i.e., graphs where one edge can connect more than two vertices simultaneously. More precisely, we use state diagrams $S = (V_S, E_S)$ that are directed hypergraphs with labelled hyperedges. Any tensor network can be translated into such a state diagram by translating the bond indices into vertices and the tensor elements into labeled hyperedges [17]. As an example, consider the TTN in Fig. 1a). It can be recast into the state diagram shown in Fig. 1b). The vertices in $V_S$ are the black dots, while the hyperedges in $E_S$ are the lines connecting the black dots. The label of every hyperedge corresponds to one tensor element and is shown by the rectangles. To avoid cluttering, we assumed that the tensors in Fig. 1a) have few non-zero elements.

Since we consider tree tensor networks, the equivalent state diagrams will have an implicit tree structure $D = (\mathcal{V}_D, \mathcal{E}_D)$, such that

1. $w \subset V_S$ for $w \in \mathcal{E}_D$,

2. $\varepsilon \subset E_S$ for $\varepsilon \in \mathcal{V}_D$,

3. $\mathcal{E}_D$ and $\mathcal{V}_D$ form a disjoint covering of $V_S$ and $E_S$, respectively.

We provide this structure for the example state diagram in Fig. 1c). The edges $w$ of $D$ are the subsets of vertices $V_S$ encircled by dashed green ellipses, and the nodes $\varepsilon$ of $D$ are the subsets of hyperedges $E_S$ encircled by dotted purple ellipses. There exists a clear one-to-one correspondence between $\mathcal{E}_D$ and $\mathcal{V}_D$ of the state diagram tree structure $D$, and $E_T$ and $V_T$ of the original TTN, respectively. Every collection of edges $\varepsilon \in \mathcal{V}_D$ corresponds to a node $s \in V_T$. As an example, consider the subset of hyperedges in Fig. 1c) denoted by $\varepsilon_B$. This subset corresponds to the tensor $B$ in Fig. 1a) and includes all of its non-zero elements as labels. Likewise, every collection of vertices $w \in \mathcal{E}_D$ corresponds to an edge $e \in E_T$. Therefore, we will usually write $\varepsilon_s$ and $w_e$. For example, the subset of vertices denoted by $w_{A,B}$ in Fig. 1c) corresponds to the edge $(A,B)$ in Fig. 1a). To avoid confusion, we will refer to $s \in V_T$ as nodes or sites when describing a quantum system and $v \in V_S$ as vertices.

Another important concept is paths through a state diagram, which can be conceptualised as a method for traversing the state diagram. We differentiate two kinds of paths: single and full paths. Single paths only contain a single hyperedge $y \in \varepsilon$ for every $\varepsilon \in \mathcal{V}_D$. Therefore, they only contain one $v \in w$ for every $w \in \mathcal{E}_D$. In the case of a single path, we include exactly one hyperedge $y_r \in \varepsilon_r$ corresponding to the root node. Additionally, the path includes all

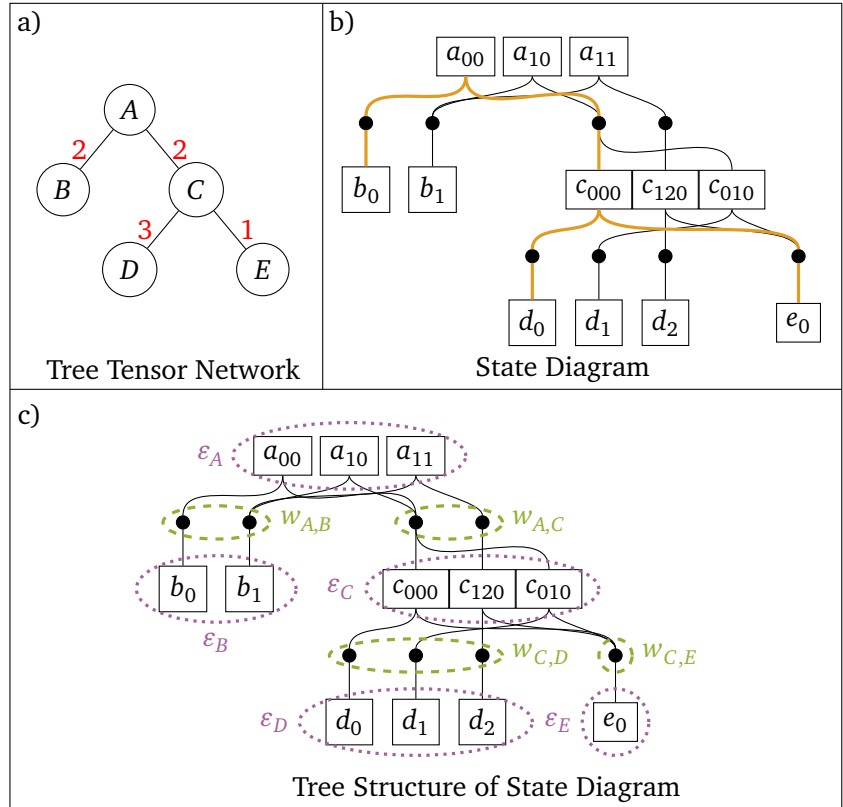

Figure 1: a) A tree tensor network with five tensors. The number next to each leg denotes the dimension of the leg. b) The state diagram corresponding to the tensor network on the left. Some tensor elements are assumed to be zero-valued and are left out to avoid cluttering. Starting at $a_{00}$ and following the single path marked in thick orange yields the term $a_{00}b_0c_{000}d_0e_0$. The full tensor network contraction can be obtained by doing this for all paths in the state diagram. c) The tree structure emerging in the state diagram. All hyperedges(/labels) belonging to one site $s$ are enclosed by dotted purple ellipses representing $\varepsilon_s$, and all vertices belonging to one edge $e$ are enclosed in dashed green ellipses $w_e$.

vertices $v_{(r,c)}$ connected to $y_r$, where $c$ are the children of $r$. Each of these vertices is also connected to hyperedges in a collection $w_c$. For every $v_{(r,c)}$ already included in the path, we include one hyperedge in $w_c$ connected to it. We repeat this process recursively until we reach the hyperedges corresponding to leafs. If we construct a full path, we include all hyperedges in one $\varepsilon$ connecting $v$. We then continue for all vertices connected by any of the hyperedges. Once more, we terminate when the leafs are reached. Constructed this way, the labels of the hyperedges represent terms of the tensor network contraction. In the case of a single path, we obtain a single term, while for a full path, we can obtain multiple terms. A single path is shown in the example state diagram in Fig. 1b) as a thick orange path through the state diagram $S$.

## 4    Finding a TTNO

A many-body quantum Hamiltonian acting on a quantum system of $L$ sites can be written as a sum of $N$-many tensor products of single-site operators $H = \sum_{j=0}^{N-1} \bigotimes_{s=0}^{L-1} A_j^{[s]}$, where $A_j^{[s]}$ is an operator acting on site $s$. Given a tree $T = (V_T, E_T)$, we can express the Hamiltonian as

$$H = \sum_{j=0}^{N-1} \underbrace{\bigotimes_{s \in V_T} A_j^{[s]}}_{h_j} . \tag{3}$$

We want to find the TTNO representation of $H$ with the same tree structure $T$. We also assume $h_j \neq h_i$ for $j \neq i$.

### 4.1    One term

---
**Algorithm 1** Create single term state diagram
---
1:  Initialize empty state diagram $D = (\mathcal{V}_D, \mathcal{E}_D)$
2:  **for** $s \in V_T$ **do**
3:      **for** $e \in E_T(s)$ **do**
4:          **if** No vertex corresponding to $e$ is in $\mathcal{V}_D$ **then**
5:              Initialize vertex $v_e$
6:              Add $v_e$ to $V_D$
7:          **end if**
8:      **end for**
9:      Initialize hyperedge $y_s$ connecting all $v_e$ for $e \in E_T(s)$
10:     Label $y_s$ with $A^{[s]}$
11:     Add $y_s$ to $\mathcal{E}_D$
12: **end for**
---

If $H = h = \otimes_{s \in V_T} A^{[s]}$ consists of only one term, the TTNO is trivial: The tensor $h^{[s]}$ of site $s$ contains $A^{[s]}$ as its only element and has as many trivial legs, i.e., bond dimension 1, as $s$ has neighbors. Similarly, it is also easy to construct the state diagram corresponding to a single term. Start with an empty state diagram $D = (\mathcal{V}_D = \emptyset, \mathcal{E}_D = \emptyset)$. Then, run through all sites $s \in V_T$. For every edge $e \in E_T(s)$ connecting to $s$ check if a vertex $v_e$ corresponding to $e$ is already in $\mathcal{V}_D$. If not create and add it to $\mathcal{V}_D$. Next, connect all vertices $v_e$ for $e \in E_T$ via a single hyperedge $y_s$ and label $y_e$ with the operator $A^{[s]}$. Add $y$ to $\mathcal{E}_D$. This algorithm is represented in Alg. 1. We give examples of single-term state diagrams in Fig. 2. Such diagrams are needed to construct state diagrams for multi-term Hamiltonians.

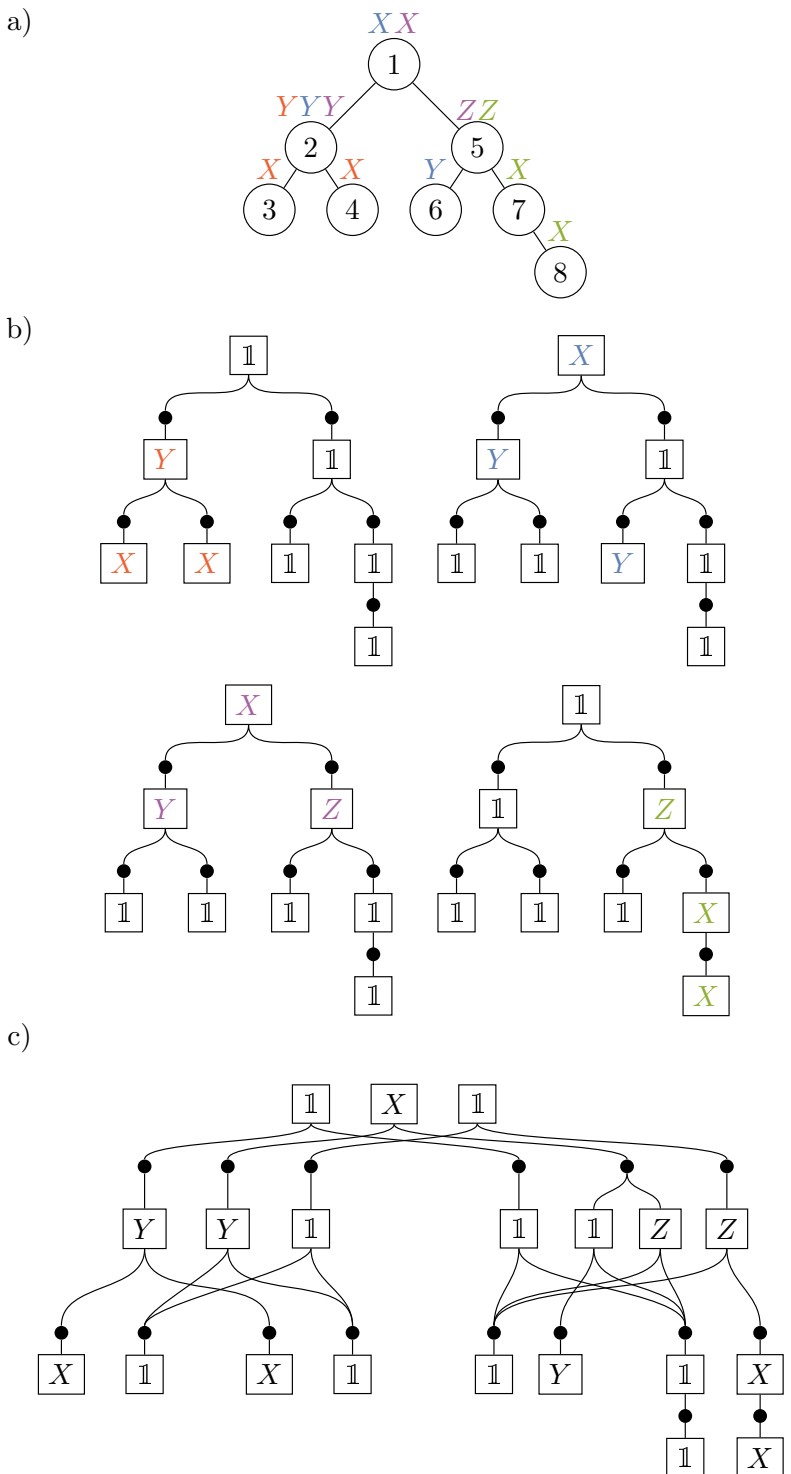

Figure 2: a) Tree structure for a given TTNO, where the Hamiltonian is given in Eq. (6). Each single site operator in the same term is given the same color and identity operators are not shown. We choose node 1 as the root of the tree structure. b) Single paths state diagrams for each term. c) Simplified full path that generates all terms.

## 4.2 Multiple terms

---

**Algorithm 2** Add term to state diagram

---

**Input:** State diagram $D = (\mathcal{V}_D, \mathcal{E}_D)$, tree tensor network $T$, new term $h_j$

1: $D_j \leftarrow$ single term state diagram $(h_j)$
2: **function** MARK_MATCHING$(J, \ell, h_j)$
3:      $L \leftarrow$ all hyperedges in $J$ with label $A_j^{[\ell]}$
4:      **for** $y \in L$ **do**
5:          **if** exist exactly one vertex $v_{\ell,s}$ connected to $y$ not marked **then**
6:              **if** $v_{\ell,s}$ is connected to only one $y \in \varepsilon_\ell$ **then**
7:                  mark $v_{\ell,s}$
8:                  $J \leftarrow$ all hyperedges in $\varepsilon_s$ connected to $v_{\ell,s}$
9:                  MARK_MATCHING$(J, s, h_j)$
10:                  break for-loop
11:              **end if**
12:          **end if**
13:      **end for**
14: **end function**
15: **for** $\ell \in \mathcal{L}(T)$ **do**
16:      $J \leftarrow \varepsilon_\ell$
17:      MARK_MATCHING$(J, \ell, h_j)$
18: **end for**
19: **for** $s \in V_T$ **do**
20:      $L \leftarrow$ all $s' \sim s$ s.t. no vertex $v \in w_{s,s'}$ marked
21:      **for** $s' \in L$ **do**
22:          add new vertex $v_{s,s'}$ to $w_{\ell,s}$
23:          mark $v_{s,s'}$
24:      **end for**
25:      $K \leftarrow$ all $y \in \varepsilon_s$ s.t. all vertices connected to it are marked
26:      **if** no $y \in K$ has label $A_j^{[s]}$ **then**
27:          add new $y$ to $\varepsilon_s$ with label $A_j^{[s]}$ connected to marked vertices
28:      **end if**
29: **end for**

---

To construct a state diagram containing more than one term, we could repeatedly add new paths to an existing $D$ obtained from Algorithm 1. Each additional term requires exactly one new path. When adding a term $h_j$, one could simply take the corresponding state diagram $D_j$ and add it to $D$ without connecting existing vertices. However, this leads to unnecessarily high bond dimensions in the TTNO. Instead, one can check if the state diagram $D$ already contains elements of $h_j$. We only want to add one path at a time, so it suffices to check if a subtree of $D$ matches with the subset of operators in $h_j$ corresponding to the same nodes. Since any subtree has to contain at least one leaf, we can define an algorithm that traverses subtrees of $D$ starting from its leaves. For each leaf $\ell \in \mathcal{L}(T)$, we check whether the corresponding set of hyperedges $\varepsilon_{s_\ell}$ in $D$ contains a hyperedge $y$ with the same label $A_j^{[s_\ell]}$ as the corresponding operator in $h_j$. In the case that the labels match, we have to check if $y$ is the only hyperedge connected to the vertex $v \in w_{(s_\ell, s_p)}$, where $s_p$ is the parent of site $s_\ell$. If there are multiple, we would add more than one path to our state diagram when connecting $v$ to the rest of the new path. There is a maximum of one hypergraph $e'$ for which both assertions are true. If there were two, we could merge them without adding or removing a path. We mark the vertex $v'$ connected to

$e'$ and move to the leaf's parent site $s_p$. There, we check if any of the hyperedges $y_{s_p} \in \varepsilon_{s_p}$ connected to $v'$ have the label $A_j^{[s_p]}$. If so, we check if all but one vertex $\tilde{v}$ connected to it are marked. If all are, the operators in $h_j$ corresponding to the subtree rooted at $s_p$ are already contained in $D$. To avoid adding more than one path, we check that $y_{s_p}$ is the only hyperedge in $\varepsilon_{s_p}$ connected to $\tilde{v}$. If so, we mark $\tilde{v}$ and continue the same procedure with the hyperedges connected to $\tilde{v}$ that is not $\varepsilon_{s_p}$. If any checks are false, we start anew with the next leaf. Once we have repeated this procedure for all leaves, we need to include the additional vertices and hyperedges required for the new path that does not yet exist in $D$. For this, we can then run over all $s \in V_T$ in arbitrary order. For each $s$, we check if there is a hyperedge $y \in \varepsilon_s$ such that all vertices connected to it are marked. If the label of $y$ is $A_j^{[s]}$, we are done. Otherwise, we create a new hyperedge with that label and connect it to all marked vertices. If not all vertices are marked, then there exists $s'$ such that $s' \sim s$ and $\varepsilon_{s'}$ contains no marked vertices. Thus, a new vertex is created, added to $\varepsilon_{s'}$ and marked. Then, we create a new hyperedge with label $A_j^{[s]}$ and connect it to all vertices in $\bigcup_{s' \sim s} \varepsilon_{s'}$ that are marked. This procedure is summarized in Alg. 2; repeating it for every term in the Hamiltonian yields a state diagram corresponding to the total Hamiltonian $H$.

From this description, we can clearly see that the algorithm scales linearly in $N$, the number of terms of the Hamiltonian. Furthermore, there is an upper bound on the scaling by considering a worst-case scenario. Our worst-case scenario occurs if, for every term, we reach the root for every search starting from a leaf. In this case, the entire algorithm would scale linearly in $|\mathcal{L}(T)|$, the number of leaves, and the tree depth depth$(T)$. Therefore, the runtime scaling of our algorithm is upper bounded by

$$\mathcal{O}(N \, |\mathcal{L}(T)| \, \mathrm{depth}(T)). \tag{4}$$

This is below the complexity of any tensor network algorithm in which we might use a TTNO.

### 4.3 State diagram to TTNO

Once a state diagram corresponding to a Hamiltonian is obtained, one can read off the TTNO tensors. For every $(s, s') \in E_T$, we find a bijection $f_{(s,s')} : w_{(s,s')} \longrightarrow \{0, \ldots, |w_{(s,s')}| - 1\}$. In this way, we assign an index value to every vertex in the state diagram $D$. This also means the bond dimension in the resulting TTNO is equal to the number of vertices at the corresponding edge. Denoting the tensor of the TTNO at site $s$ as $h^{[s]}$, we define its elements by running through the hyperedges $y_s \in \varepsilon_s$. Let $\sigma$ be an arbitrary ordering of $E_T(s)$, the edges connected to site $s$. We define the tensor elements as

$$h^{[s]}_{f(v_{\sigma_1}), f(v_{\sigma_2}), \ldots, f(v_{\sigma_{|E_T(s)|}})} = A^{[s]}, \tag{5}$$

where $\{v_i\}$ are the vertices connected to a hyperedge $y_s$ and $A^{[s]}$ is the hyperedges's label. Doing this for all sites and hyperedges yields all tensors in the TTNO, which is equivalent to the original Hamiltonian.

## 5 Examples

Now, we have a way to convert a given Hamiltonian into a state diagram and can translate that state diagram into a TTNO. To improve the understanding of the algorithm 2 we will look at some examples.

## 5.1 Toy examples

As an initial example, we look at a toy model. We define a Hamiltonian acting on a given tree structure where the set of single-site operators $\{A_j\} = \{\mathbb{1}, X, Y, Z\}$. For concreteness, we take these to be the Pauli operators. However, it is only important that they are all different. Our example Hamiltonian will act on a total system consisting of eight sites with an underlying tree structure as given in Fig. 2 a) with node 1 as the root. For this example, we define the Hamiltonian specifically as

$$H_{\text{toy}} = \sum_{j=1}^{4} h_j = Y_2 X_3 X_4 + X_1 Y_2 Y_6 + X_1 Y_2 Z_5 + Z_5 X_7 X_8 \,. \tag{6}$$

In Fig. 2b) we constructed the state diagrams corresponding to each of the four terms in (6). Every diagram has seven vertices, one for each bond, and eight hyperedges, one for each site. We can use our algorithm to individually add the state diagrams corresponding to terms $h_2, h_3$, and $h_4$ to the diagram corresponding to $h_1$. As a result, we obtain the large state diagram in Fig. 2c). We can immediately notice that no edge has more than three vertices corresponding to it. The naive solution of combining the diagrams without connecting them would result in four vertices at every edge. Since the number of vertices is the number of bond dimensions in the resulting TTNO, this is an advantage. The choice of the root does not influence the size of the bond dimensions in the resulting TTNO, with one exception. Choosing a site with only one neighbor as the root results in a significantly worse performance of our algorithm. This can be traced back to the fact that a site with only one neighbor would be a leaf if it were not the root. To showcase this, we give the resulting state diagrams for two different root choices in App. A.

For sufficiently small systems, we can construct the complete matrix presentation of a Hamiltonian. Given the Hamiltonian as a matrix, we obtain a TTNO (as a reference for comparison) by using singular value decompositions (SVD). The resulting TTNO will then have the minimal possible bond dimension due to the truncation of zero-valued singular values. To benchmark our algorithm in terms of small bond dimensions, we run it for random Hamiltonians on the tree structure in Fig. 2a) with the same four single-site operators. Additionally, we vary the number of terms of the Hamiltonian. Some of the results are shown in Fig. 3.

Notice that our algorithm does not always find the optimal bond dimension. In Fig. 3a we can clearly see that many data points are below the blue line. The blue line visualizes where the bond dimension found via our algorithm 2 would be equal to the bond dimension found by using SVDs. However, we can see that the darkest points, i.e., the ones with the most samples are on the blue line. Furthermore, most points are still close to optimality.

A related question concerns the difference in the bond dimension obtained by our algorithm compared to the optimum, depending on the number of terms in the Hamiltonian. Thus, we also plot this difference after averaging over all bonds:

$$r_{\text{diff}} = \frac{\sum D_{\text{alg}} - D_{\text{svd}}}{N_{\text{samples}} N_{\text{bonds}}} \,. \tag{7}$$

Here, we sum over all bond dimensions we obtained in the numerical experiment. We find that the more terms the Hamiltonian is made of, the higher $r_{\text{diff}}$. While our algorithm does not provide the optimal bond dimensions, it is an efficient way to obtain a Hamiltonian in TTNO form without ever creating a high-degree tensor or a high-dimensional matrix. Once the TTNO form is obtained, the optimal dimension may be found by SVDs. Since the cost of a single SVD sweep is usually low compared to the total tensor network algorithm and the TTNO is reused many times, this is a feasible way to obtain the optimal TTNO. Instead of a normal SVD one can also use the compression methods for MPOs proposed in [36], which are easily

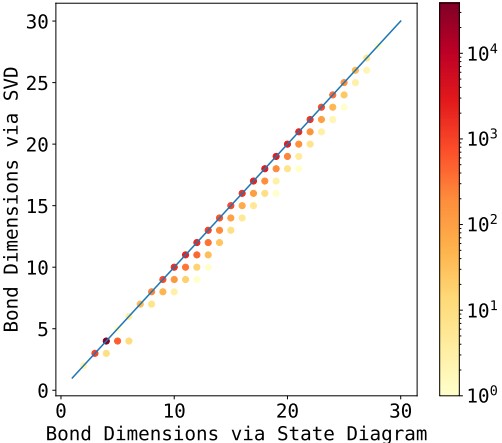
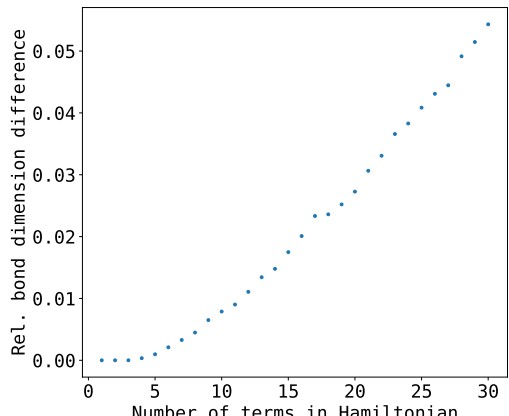

(a) The bond dimension for 10000 random sample Hamiltonians with 30 terms obtained via our algorithm versus the optimal (minimal) bond dimension based on singular value decompositions. A darker colour represents more of our sample bonds with the given relation of found to optimal bond dimension. The blue line shows $y = x$.

(b) The number of terms in the Hamiltonian against the average difference in bond dimension as obtained via our algorithm compared to the minimal dimension per bond.

Figure 3: Plots visualising the difference of the bond dimension as found by our algorithm compared to the optimal bond dimension that can be achieved using singular value decompositions.

generalizable to TTNO. They consider sparsity and the appearance of vastly different singular values.

## 5.2 Nearest neighbour interactions

While nearest neighbor interactions have already been thoroughly investigated on Cayley trees [37, 38], we can find the TTNO representing nearest neighbour interactions on an arbitrary tree structure $T$. Such interactions are given by a Hamiltonian of the form

$$H_{\mathrm{nn}} = \sum_{s \sim s'} A^{[s]} A^{[s']}, \qquad (8)$$

where an operator $A^{[s]}$ is applied to the site $s$. Note that we assume all operators $A$ to be different in every term. We do not make this explicit in the notation to avoid clutter. Assuming the tree $T$ is rooted in a vertex $r \in V_T$, we can rewrite (8) as

$$H_{\mathrm{nn}} = \sum_{s \text{ s.t. } s \sim r} A^{[r]} A^{[s]} + \sum_{r \sim s} \left( \sum_{s \sim c} A^{[s]} B^{[c]} + \mathbb{1}^{[s]} C \right), \qquad (9)$$

where all terms that act non-trivially only on the subtree rooted in $s$ and couple to $A^{[s]}$ are denoted by $B^{[c]}$. All terms that are additionally trivial on $s$ are collected in the operators $C$. Thus, we can combine all of the tree's vertices that are children of a given root child $s$ into a single vertex. Using our algorithm, we can find a state diagram as pictured in Fig. 4 representing the rewritten Hamiltonian (9).

Let us take a closer look at the terms involving the first child $s_1$ of the root $r$. All terms that act non-trivially only on the subtree below $s_1$ are connected via the vertex indexed by 2.

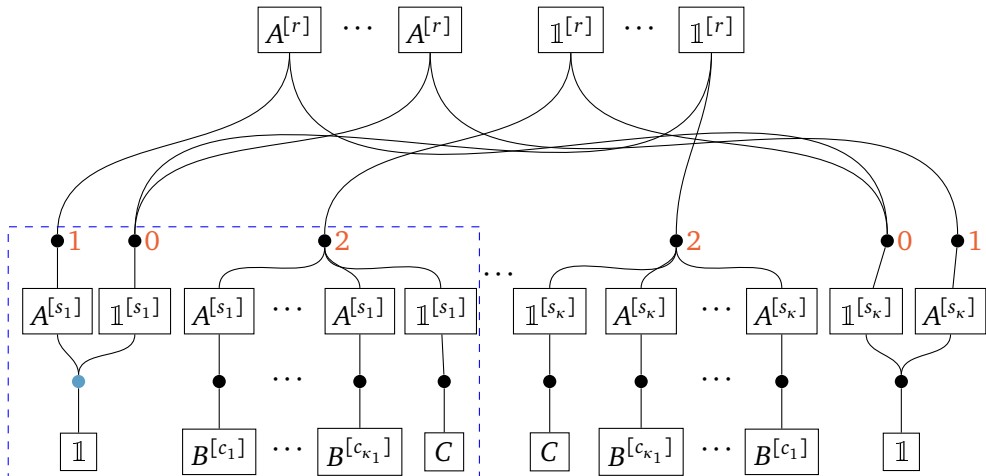

Figure 4: The state diagram corresponding to the rewritten Hamiltonian (9) on an arbitrary tree structure. $r$ has $\kappa$ children, while the $j$th child $s_j$ has $\kappa_j$ children. The dashed blue box contains the parts of the state diagram corresponding to the subtree rooted at vertex $s_1$. All terms acting non-trivially only in that subtree are rooted to the vertex on the left indexed by 2. When recursively building the state diagram, we can reuse the vertex marked in cyan to minimize the bond dimensions. The red digits correspond to the index values of the TTNO-tensor at the root $r$.

Considering the algorithm Alg. 2, this happens since the subtree above this vertex has only identities acting on it for every term. Thus, it suffices to have a single path through the remaining tree consisting of identities. Conversely, all terms that act trivially on the subtree of $s_1$ are connected via the vertex indexed by 0. The only term that acts non-trivially on both subtrees is the interaction between $r$ and $s_1$ and can be connected via the vertex indexed as 1. There is an analogy between the total tree and the terms acting trivially only on the subtree below $s_1$. By considering $s_1$ as the root of this subtree and forgetting about the remaining tree, we can recursively build the complete state diagram. Notably, we can reuse the vertex marked in cyan for terms acting trivially on $s_1$ itself. Thus, we obtain the same state diagram structure below $s_1$. We repeat this procedure recursively and for all children of the root $r$.

Thus, we find the TTNO tensor elements at a site $s$ to be

$$
h^{[s]}_{i_1,\dots,i_{\kappa_s},i_p} = \begin{cases} \mathbb{1}, & i_j = 0, \ \forall j, \\ A_p, & i_p = 1 \wedge i_{j'} = 0, \ \text{for } j' \neq p, \\ A_j, & i_j = 1 \wedge i_p = 2 \wedge i_{j'} = 0, \ \text{for } j' \notin \{j,p\}, \\ \mathbb{1}, & i_j = i_p = 2 \wedge i_{j'} = 0, \ \text{for } j' \notin \{j,p\}, \\ 0, & \text{else}, \end{cases} \tag{10}
$$

where $A_j$ is the operator applied to site $s$ in the interaction term with site $j$. Additionally, $p$ denotes the parent site of $s$ and indices $\{1\dots\kappa_s\}$ denote the $\kappa_s$ children of $s$. We can reduce this for some of the special vertices. Since the root $r$ does not have a parent, we find

$$
h^{[r]}_{i_1,\dots,i_{\kappa_s}} = \begin{cases} A_j, & i_j = 1 \wedge i_{j'} = 0, \ \text{for } j' \neq j, \\ \mathbb{1}, & i_j = 2 \wedge i_{j'} = 0, \ \text{for } j' \neq j \wedge j \text{ no leaf}, \\ 0, & \text{else}, \end{cases} \tag{11}
$$

on the other hand, since leaves do not have children, the respective tensor of a leaf $\ell$ reduces

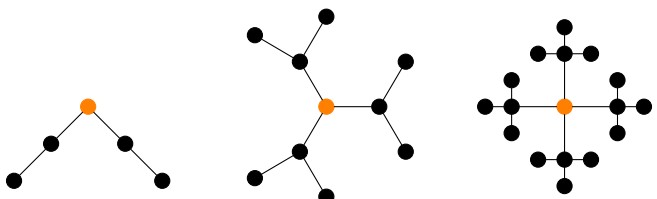

Figure 5: Full Cayley trees of depth $D = 2$ of degree $\kappa = 2, 3, 4$ (from left to right). The root is colored in orange.

to

$$
h_{i_p}^{[\ell]} = \begin{cases} \mathbb{1}, & i_p = 0, \\ A_p, & i_p = 1, \\ 0, & \text{else}. \end{cases} \tag{12}
$$

This also implies that the parents of leaves have a smaller bond dimension to these leaves. Consequently, for nearest-neighbor interactions, the bond dimension is independent of system size. However, the number of elements required in the TTNO-tensor of a site scales exponentially with the number of children.

Some Hamiltonians, such as the Ising model [39], contain single site operators $Z_s$ in addition to the nearest neighbor interaction. Such terms can be incorporated in the above tensors by changing a single element:

$$
h_{0,\ldots,0}^{[r]} = Z_r, \qquad \text{for the root } r, \tag{13}
$$

$$
h_2^{[\ell]} = Z_\ell, \qquad \text{for all } \ell \in \mathcal{L}(T), \tag{14}
$$

$$
h_{0,\ldots,0,2}^{[s]} = Z_s, \qquad \text{for all other } s \in V_T. \tag{15}
$$

## 5.3 Long-range interactions

For long-range interactions, we restrict our trees to be full Cayley trees of degree $\kappa$. This means every node, except for the leaves, is connected to $\kappa$ other nodes and there exists a node $r \in V_T$ such that for all leaves $\ell \in \mathcal{L}(T)$ the distance $d(r,s) = D$ for a fixed $D \in \mathbb{N}$. See Fig. 5 for some examples. We call $D$ the depth of the tree and choose $r$ as its root. For a given interaction range $\chi$, we want to find the maximum bond dimension required to represent the Hamiltonian

$$
H_\chi = \sum_{(s,s') \in M_\chi} A_{s'}^{[s]} A_s^{[s']}, \tag{16}
$$

where $M_\chi = \{(s,s') \in V_T \times V_T \mid d(s,s') = \chi \wedge s < s'\}$ for an arbitrary ordering $<$ of the vertices. The operators acting on the same site are in general different in each term. Therefore, we know that there will not be any equal subtrees in the single-site diagrams except for the trivial subtrees where all sites are acted upon by $\mathbb{1}$. Therefore, the dimension of a bond equivalent to an edge $(v, v') \in E_T$ increases by one for every term $A_{s'}^{[s]} A_s^{[s']}$, such that $(v, v') \in E_{\gamma(s,s')}$. The bond dimensions of the root tensor are maximal in this situation, so we determine it as an upper bound for the rest of the network. With the argumentation above, we need to determine for each child $s$ of $r$ the number of pairs $(v, v') \in V_T \times V_T$ such that $(s, r) \in E_{\gamma(v,v')}$. We immediately know that $v$ and $v'$ have to be in the subtree of different children of $r$. We will at first consider the number of pairs from one such subtree $T_{|s}$ to a different one $T_{|s'}$. The

following identity will be useful

$$\left| T_{|s} \cap \partial \mathcal{B}_R(r) \right| = (\kappa - 1)^{R-1}, \tag{17}$$

for any $R \in \mathbb{N}$ such that $\leq 2D - 1$ and where $|A|$ for a subset $A \subset T$ denotes the number of nodes in $A$. For any $c \in T_{|s}$ with $\delta = d(c, r)$ we find

$$\left| T_{|s'} \cap \partial \mathcal{B}_\chi(c) \right| = \left| T_{|s'} \cap \partial \mathcal{B}_{\chi-\delta}(r) \right| = (\kappa - 1)^{\chi - \delta - 1}. \tag{18}$$

There exist

$$\left| T_{|s'} \cap \partial \mathcal{B}_\delta(r) \right| = (\kappa - 1)^{\delta-1}, \tag{19}$$

many different $c$. Therefore if $\chi \leq D$ the total number of pairs $(c, c') \in T_{|s} \times T_{|s'}$ fulfilling the above condition are

$$\sum_{\delta=1}^{\chi-1} (\kappa - 1)^{\delta-1} (\kappa - 1)^{\chi-\delta-1} = (\chi - 1)(\kappa - 1)^{\chi-2}. \tag{20}$$

Additionally, we have one term for each interaction between $r$ and sites in $T_{|s}$ and consider the two bond dimensions required to represent the trivial subtree on either side of the edge $(r, s)$. Considering that there are $\kappa$ children of $r$ and that the tree is symmetric around $r$, we obtain the total maximum required bond dimension

$$(\kappa - 1)(\chi - 1)(\kappa - 1)^{\chi-2} + (\kappa - 1)^{\chi-1} + 2 = 2 + \chi(\kappa - 1)^{\chi-1}. \tag{21}$$

Note that for the case of an MPO, i.e., $\kappa = 2$, we obtain the well-known linear scaling of the bond dimension with system size. In the case of $\chi > D$, we have to replace Eq. (20) by

$$\sum_{\delta=\chi-D}^{D} (\kappa - 1)^{\delta-1} (\kappa - 1)^{\chi-\delta-1} = (2D - \chi)(\kappa - 1)^{\chi-2} \theta(\chi - 2D), \tag{22}$$

where $\theta$ represents the Heaviside step function.

If we have an all-to-all connectivity instead of a fixed interaction length, the Hamiltonian reads as

$$H = \sum_{\chi=0}^{2D-1} \sum_{(s,s') \in M_\chi} A_{s'}^{[s]} A_s^{[s']}. \tag{23}$$

Using our previous results and assuming $\kappa > 2$, we obtain the bond dimensions of the tensor at $r$ as

$$2 + \sum_{\chi=1}^{D} \chi(\kappa - 1)^{\chi-1} + \sum_{\chi=D+1}^{2D-1} (2D - \chi)(\kappa - 1)^{\chi-1}, \tag{24}$$

which scales as $\mathcal{O}((\kappa - 1)^{2(D-1)})$. However, the number of sites in the tree is given by

$$|V_T| = 1 + \sum_{\Delta=1}^{D} \kappa(\kappa - 1)^{D-1}, \tag{25}$$

for $D \geq 1$. Thus, the maximum bond dimension scales linearly $\mathcal{O}(|V_T|)$ in the number of sites.

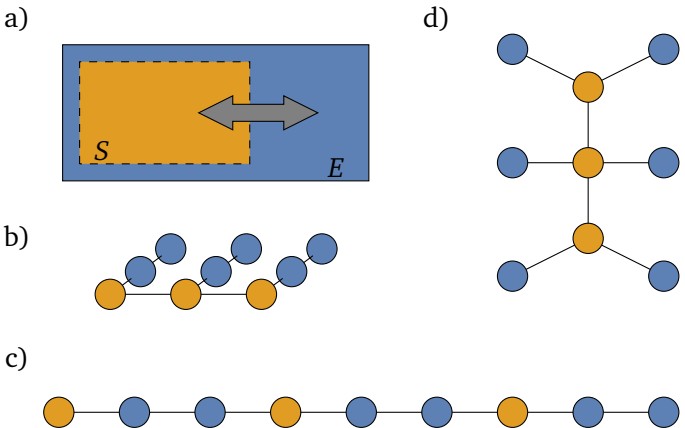

Figure 6: a) Schematic of an open quantum system. A principal system $S$ interacts with an environment $E$. b)–d) The three different tree structures that we consider. b) The fork tensor product network, c) the star tree network, d) and the matrix product/chain network. The orange nodes correspond to the spin sites and the blue ones to the bosonic bath sites.

## 6 Application to open quantum systems

We will now consider an application well suited for the use of TTN inspired by open quantum systems. In open quantum systems, the Hamiltonian generally has the form

$$H = H_S + H_E + H_{SE}.\tag{26}$$

It is split into three parts: The Hamiltonian of the principal system $H_S$, which can be interpreted as the experimentally accessible system, the Hamiltonian of the environment $H_E$, and the interaction between the two mediated by $H_{SE}$. A sketch of this concept is also shown in Fig. 6a). As a principal system, we will consider a spin-$\frac{1}{2}$ Heisenberg chain of length $N$:

$$H_S = -J \sum_{s=0}^{N-2} \langle \vec{\sigma}, \vec{\sigma} \rangle,\tag{27}$$

where $J \in \mathbb{R}^+$ is the interaction strength and $\vec{\sigma} = \begin{pmatrix} X & Y & Z \end{pmatrix}^T$ is the vector of Pauli operators. Every spin is coupled to $M$ bosonic environment sites via

$$H_{SE} = \sum_{s=0}^{N-1} \sum_{b=0}^{M-1} \left( g_{s,b} Z_s B_{s,b} + \text{h.c.} \right),\tag{28}$$

where $g_{s,b}$ is the coupling strength of spin $s$ with boson $(s,b)$ and $B$ and $B^\dagger$ are the bosonic annihilation and creation operators. We assume that the bosons do not interact with each other. Therefore, the environment Hamiltonian is given by

$$H_E = \sum_{s=0}^{N-1} \sum_{b=0}^{M-1} \omega_{s,b} \mathcal{N}_{s,b},\tag{29}$$

where $\omega_{s,b}$ is the characteristic frequency of a boson and $\mathcal{N}$ the bosonic number operator. For simplicity, we assume a homogenous model with $\omega = \omega_{s,b}$ and $g = g_{s,b}$ for all $(s,b)$.

The interaction structure makes this problem suited to a tree topology. We will run our algorithm on three different tree topologies, a depiction of which is given in Fig. 6b–d). The first topology is a chain as found in Fig. 6c). The leftmost tensor represents a spin site. It is followed by all the tensors representing the bosonic sites coupled to the first spin. The last bosonic site is then connected to the next spin site. Therefore, we are just searching for an MPO representation of the interaction. Our algorithm finds bond dimensions of $(5, 6)$ for the non-boundary spin sites as well as $(6, 6)$ or $(6, 5)$ for the non-boundary bosonic sites, depending on their position in the chain. The bond dimension for the boundary sites is lower, as expected. One can find the same bond dimension and in fact, the same tensors up to row and column order using the finite automaton method for MPOs [12]. A choice of the explicit tensors can be found in Appendix B. For the fork tensor product (FTP) [1] network all the bosons coupling to the same spin site are placed in a chain. The chain is attached to the spin site at one of its ends. Additionally, the spins are attached to each other in an additional chain. Using our algorithm, we find that the tensors of the spin sites have bond dimensions of $(5, 5, 3)$, while the bosonic sites' are reduced to $(3, 3)$. Notably, the number of elements required to represent the TTNO with an FTP topology is smaller for $N > 3$ and arbitrary $M$ than the number of elements required to represent the equivalent MPO. This shows the advantage of using a tree tensor network over one-dimensional chains for specific problems. This difference becomes especially important if the operator is applied to a quantum state many times, as is the case in time-evolution methods [19, 24] or stochastic methods for open quantum systems [22, 40]. However, this advantage does not carry over to the star-shaped topology. Here, we find a bond dimension for the spin sites of 5 for the bonds connecting to other spins and of 3 for the bonds connecting to bosonic sites. However, since we have to add an additional leg for each additional bosonic site, the number of elements required to represent each spin tensor scales exponentially in $N$.

## 7 Discussion and future work

Using state diagrams as a data structure, we found a way to construct TTNO for a given Hamiltonian. Notably, the algorithm can find a TTNO close to the optimal bond dimension. However, there is still some compression required to reach the optimal dimension. It would, therefore, be interesting to see if one can combine concepts from our algorithm and the construction of MPO using bipartite graph theory [14] to find an optimal construction scheme for arbitrary TTNO. We also found that using tree topologies and TTNO can be better than just using a one-dimensional chain with MPOs. However, one can observe, as is also usual for physics-inspired MPOs, that the tensors are very sparse. Therefore, one could continue the exploration of TTN and state diagrams by finding efficient ways to directly apply the state diagram to a TTNS. There might also be modifications that can be made to the DMRG and TDVP methods, which allow a direct application of state diagrams, thus improving their performance when used on TTN. As a final direction for future work, one could try to conceive algorithms that find an optimal tree topology with respect to the total number of elements or bond dimension.

## Acknowledgments

The authors want to thank the International Quantum Tensor Network for the opportunity to discuss our findings with the Tensor Network Community.

**Funding information** The research is part of the Munich Quantum Valley, which is supported by the Bavarian state government with funds from the Hightech Agenda Bayern Plus. The research is also supported by the Bavarian Ministry of Economic Affairs, Regional Development and Energy via the project BayQS with funds from the Hightech Agenda Bayern.

# A   Influence of root node choice

In section 5.1 we always used the same tree structure and labelled the first node as the root. In this appendix, we show the impact of choosing a different node as the root. First, we choose node 5 as the root for the tree in Fig. 1a). Repeating the process to obtain the state diagram for the toy Hamiltonian (6), we arrive at the diagram in Fig. 7. This state diagram is just a shifted version of the diagram where the root node is node 1 seen in Fig. 1c). One can find that the output of Alg. 2 is independent of the choice of root as long as the root is not a leaf. This is further supported by the results in Fig. 9a, showing the bond dimensions for 10000 different Hamiltonians defined on the tree Fig. 1a). The bond dimensions found by our algorithm are plotted against those found via SVDs. Fig. 9a is exactly the same as Fig. 3a showing the data for the root at node 1.

However, there is a significant change in the algorithm's performance if we choose a leaf node as the root. For example, choose node 6 in Fig. 1a) as the root. For this case the state diagram obtained by applying the algorithm to (6) is given in Fig. 8. Comparing this to the original case with the root at node 1 shown in Fig. 1c), we find that the bond dimension between nodes 5 and 6 grows from 3 to 4, while the others do not change. We observe a similar result if we compute the bond dimensions for many random Hamiltonians. While many of the bond dimensions are still close to the diagonal, the points spread further. Additionally, we get a group of data points where the bond dimension found via Alg. 2 is far higher than their counterpart found via SVDs. These are likely the dimensions of the bond between nodes 5 and 6. Overall, our numerical findings suggest that the impact of choosing a leaf as the tree structure's root significantly harms the performance of Alg. 2. On the other hand, as expected, the results are independent of the choice of the root as long as it is not a leaf node.

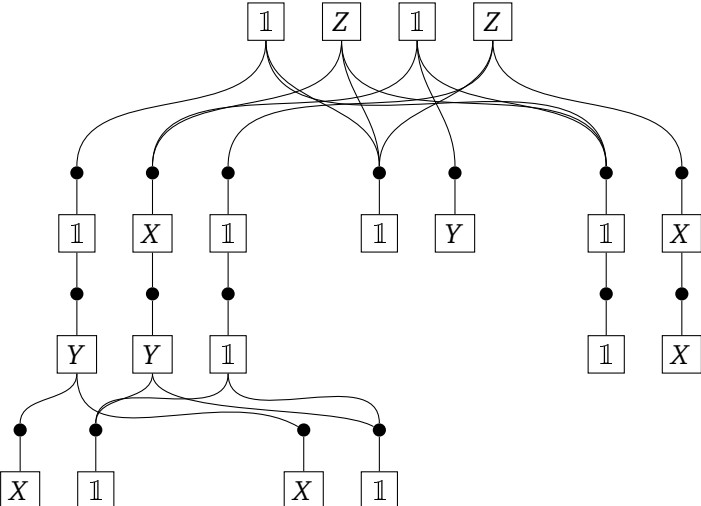

Figure 7: The state diagram of the toy Hamiltonian (6), if we choose node 5 to be the root of the tree structure.

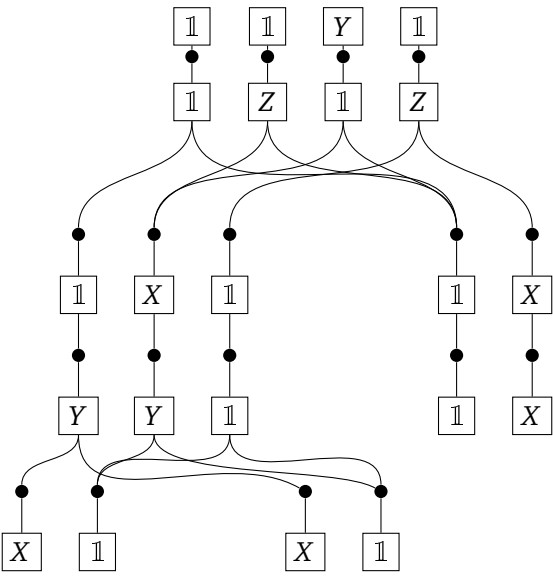

Figure 8: The state diagram of the toy Hamiltonian (6), if we choose node 6 to be the root of the tree structure.

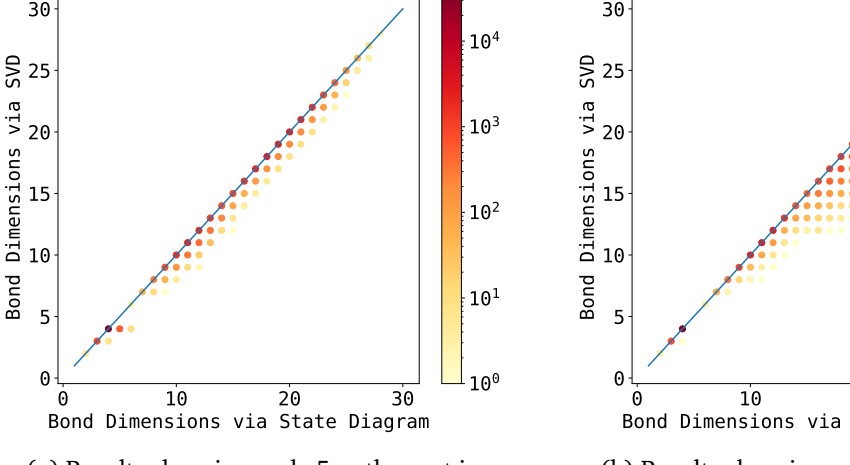

(a) Results choosing node 5 as the root in the tree from Fig. 1a).

(b) Results choosing node 6 as the root in the tree from Fig. 1a).

Figure 9: Plots showing the bond dimension for random Hamiltonians with 30 terms obtained via our algorithm versus the optimal (minimal) bond dimension based on singular value decompositions. For both the roots are different compared to the example in section 5.1. The blue line shows $y = x$.

## B  Open quantum system MPO

This appendix provides the explicit form of the MPO tensors corresponding to (26). The spin tensors are given by the operator-valued matrix

$$
h^{[s]} = \begin{pmatrix}
I & 0 & 0 & 0 & 0 & 0 \\
-JX & 0 & 0 & 0 & 0 & 0 \\
-JY & 0 & 0 & 0 & 0 & 0 \\
-JZ & 0 & 0 & 0 & 0 & 0 \\
0 & Z & X & Y & Z & I
\end{pmatrix}.
\tag{B.1}
$$

For $s = 0$ we only take the last row vector, and for $s = M-1$ we leave out the columns 2, 3, and 4. For the bosons, there are two distinct matrices. The first one is only valid for every boson of the form $(s, M-1)$, so the last boson is coupled to a spin in the chain. The corresponding matrix is

$$
h^{[(s,M-1)]} = \begin{pmatrix}
I & 0 & 0 & 0 & 0 \\
gB + g^*B^\dagger & 0 & 0 & 0 & 0 \\
0 & I & 0 & 0 & 0 \\
0 & 0 & I & 0 & 0 \\
0 & 0 & 0 & I & 0 \\
\omega\mathcal{N} & 0 & 0 & 0 & I
\end{pmatrix}.
\tag{B.2}
$$

Otherwise, we find

$$
h^{[(s,b)]} = \begin{pmatrix}
I & 0 & 0 & 0 & 0 & 0 \\
gB + g^*B^\dagger & I & 0 & 0 & 0 & 0 \\
0 & 0 & I & 0 & 0 & 0 \\
0 & 0 & 0 & I & 0 & 0 \\
0 & 0 & 0 & 0 & I & 0 \\
\omega\mathcal{N} & 0 & 0 & 0 & 0 & I
\end{pmatrix}, \quad \text{for } b \in \{0, \dots, M-1\}.
\tag{B.3}
$$

Notably for $s = N-1$ we cut the rows and columns 1, 2, and 3 in (B.2) as well as 2, 3, and 4 in (B.3).

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
