# Peer review of "State Diagrams to determine Tree Tensor Network Operators"

_SciPost Physics Core, doi:SciPost Phys. Core 7, 036 (2024)_

## Round 3 · Referee Report · Anonymous (Referee 1) · 2024-5-17

Strengths
1- This paper describes a general method to form almost-optimal tree-tensor-network operators (TTNOs) for a general tree-tensor network (TTN). TTNs, which are generalizations of matrix-product states, are useful variational ansatz states for a variety of strongly correlated quantum problems including, but not limited to, quantum impurity problems (Kondo, Anderson, DMFT), post-Hartree-Fock quantum chemistry, and open quantum systems. It is therefore presents a conceptual and methodic advance in TTN technology, in particular, making it possible to use TTNOs in a general and efficient way. 2- The method, adapting state machines to the geometry and topology of TTNs so that general TTNOs that can also be long-ranged can be efficiently treated, is original and clever. 3- The paper is carefully and clearly written, and the method is described in detail, in words, in pseudocode, and in state diagrams. As such, it makes both the conceptual aspects of the method and the implementation aspects accessible to the reader.
Weaknesses
1- While the paper describes a methodic advance and indicates areas of application in which the use of efficient TTNOs will be useful, in particular, in the area of open quantum systems, no physical results are presented in the paper. The example treated for open quantum systems is the spin-1/2 Heisenberg chain coupled to a bath modelled by bosonic sites. While this is an interesting test system, it is not necessarily a system of particulary high experimental and physical interest. In addition, the model is only used as example to explore the efficacy of the TTNO representation; a calculation of the physical properties of the model, not even to show what the method is capable of, is not carried out.
Report
The general acceptance criteria for SciPost articles are certainly met by this paper. The SciPost Core-specific expectations are that the work:
1- Address an important (set of) problem(s) in the field using appropriate methods with an above-the-norm degree of originality.
I find that the authors address the somewhat technical problem of how best to form TTNOs for TTNs in an original, insightful and general way. In doing so, they make an import technical advance for the set of TTN-based methods. Tensor-network methods in general (in particular, MPS- and iPEPS-based method) are a highly useful and fast developing set of methods. TTN methods are an interesting and variant of TNS methods, but not do not (at least yet) have the same degree of usefulness and wide range of applicability as other TNS methods.
- Detail one or more new research results significantly advancing current knowledge and understanding of the field.
The new research results are of a methodic, technical nature. A such, they do make a significant advance in one technical aspect of a specific kind of tensor network. However, as detailed above under weaknesses, this work does not contain any new physical results; the authors make clear that this is not their intention.
Requested changes
1- I find that the notation in Eqs. (17), (18), and (19) is not completely defined. The authors should define all symbols and notation that are used. 2- The paper should be proofread for punctuation, especially commas and hyphenation.
Recommendation
Ask for minor revision

---

## Round 4 · Author Response

We have fixed the points brought up by the referee and are thankful for the feedback given.

---

## Round 4 · List of Changes

• Removed a short paragraph and an reference to the virtual vertices at the beginning and end of the state diagrams. After reading through the paper, we found that these concepts did not help the understanding and added to the already convoluted assortment of newly defined objects.
  • Improved the explanation of equations (17), (18), and (19). Specifically we added a definition for the |*| notation.
  • General proofreading and typo corrections.

---

## Editorial Decision

published